# The influence of receiving real-time visual feedback on breathing during treadmill running to exhaustion

Joseph A. Passafiume[1]*, Nelson A. Glover[1], Anne R. Crecelius[2], Ajit M. W. Chaudhari[1,3,4]

**1** Department of Mechanical and Aerospace Engineering, The Ohio State University, Columbus, Ohio, United States of America, **2** Department of Health and Sport Science, University of Dayton, Dayton, Ohio, United States of America, **3** School of Health and Rehabilitation Sciences, The Ohio State University, Columbus, Ohio, United States of America, **4** Sports Medicine Research Institute, The Ohio State University, Columbus, Ohio, United States of America

* passafiume.6@osu.edu

**Data Availability Statement:** All data will be available in a repository upon acceptance.

**Funding:** The authors received no specific funding for this work.

## Abstract

Breathing plays a vital role in everyday life, and specifically during exercise it provides working muscles with the oxygen necessary for optimal performance. Respiratory inductance plethysmography (RIP) monitors breathing through elastic belts around the chest and abdomen, with efficient breathing defined by synchronous chest and abdomen movement. This study examined if providing runners with visual feedback through RIP could increase breathing efficiency and thereby time to exhaustion. Thirteen recreational runners (8F, 5M) ran to exhaustion on an inclined treadmill on two days, with visual feedback provided on one randomly chosen day. Phase angle was calculated as a measure of thoraco-abdominal coordination. Time to exhaustion was not significantly increased when visual feedback was provided ($p = 1$). Phase angle was not significantly predicted by visual feedback ($p = 0.667$). Six participants improved phase angle when visual feedback was provided, four of whom increased time to exhaustion. Four participants improved phase angle by 9° or more, three of whom increased time to exhaustion. Participants who improved phase angle with visual feedback highlight that improving phase angle could increase time to exhaustion. Greater familiarization with breathing techniques and visual feedback and a different paradigm to induce running fatigue are needed to support future studies of breathing in runners.

## Introduction

Running has been established as one of the most popular forms of physical activity in the United States for several years [1]. Running, like any form of exercise, has metabolic requirements that the body must meet, and a primary way that the body meets these requirements is through breathing [2]. Breathing allows for the exchange of oxygen and carbon dioxide between the lungs and blood, and the heart pumps oxygen-carrying red blood cells to the working muscles for fuel, which allows muscle contractions to continue through the

**Competing interests:** The authors have declared that no competing interests exist.

production of adenosine triphosphate via the metabolic pathways of glycolysis and mitochondrial respiration [3–6]. Through maximum oxygen uptake testing, it is known that those who are better at exchanging oxygen with the body can perform endurance exercise for longer periods of time [7, 8].

The most efficient breathing entails the chest and abdomen expanding and contracting in synchrony, which allows for no interference with the contracting diaphragm and maximal air intake into the lungs, therefore maximizing the amount of oxygen going into the bloodstream with each breath [9–12]. The breathing technique of the chest and abdomen expanding and contracting in synchrony has been implemented in various settings previously, which ultimately showed improved oxygen intake in a population of participants with chronic obstructive pulmonary disease [9] as well as reduced oxidative stress in the bloodstream following exhaustive exercise [10]. Furthermore, on breathing in general during exercise, tidal volume has been shown to be relevant to exertion level and maximal oxygen uptake [11], and individuals with more synchronous chest and abdomen movements during a graded run experienced less oxygen desaturation [12]. Combining these aspects of efficient breathing and breathing during exercise, improving breathing patterns during exercise could lead to performance improvements through improved oxygen exchange.

This efficient breathing can be measured and displayed in real time using respiratory inductance plethysmography (RIP) [13–16], creating an opportunity to use RIP for biofeedback training. RIP is a non-invasive method of monitoring breathing through the placement of elastic belts around the chest and abdomen, referred to as RIP belts, that vary their voltage outputs as their circumference changes [17]. RIP has been shown to be reliable and accurate when compared to other measurements of breathing at rest [18, 19] and during exercise [20–24]. While there have been studies that have attempted to alter breathing pattern during exercise [25–27], there are no studies that have investigated if optimizing breathing patterns can improve running performance. Breathing for chest and abdomen synchronicity has been taught previously [28] but has not been applied in sport-based research.

Synchrony between the chest and abdomen is quantified through the absolute value of the phase angle [20, 22, 29]. Phase angle values range from 0–180°, where a value of 0° signifies perfectly synchronous movement of the chest and abdomen, shown in Fig 1. Common phase angle values of humans range from 0–24° at rest [13] and average out to approximately 20° during exercise to exhaustion [20].

Given that the body relies on breathing to carry out physical demands and that breathing in healthy individuals is typically not optimally efficient while at rest or exercising [12, 13], this study sought to examine if time to exhaustion of runners using an inclined graded running paradigm could be increased if real-time visual feedback on the breathing pattern of the chest and abdomen was provided during an exhaustive run. It was hypothesized that having visual feedback during the run would improve breathing efficiency, in turn leading to an increased time to exhaustion.

## Materials and methods

### Participant population

Thirteen adult recreational runners (eight females, five males) participated in this study, with recreational runners defined as those who run eight to thirty-two kilometers per week on average. Participant demographics are provided in Table 1. All participants did not have a previous lower limb surgery, medically diagnosed breathing deficiency, nor a running related injury in the past six months [30]. All participants filled out an initial contact survey to ensure the previously mentioned criteria were met. The methods of this study were approved by the

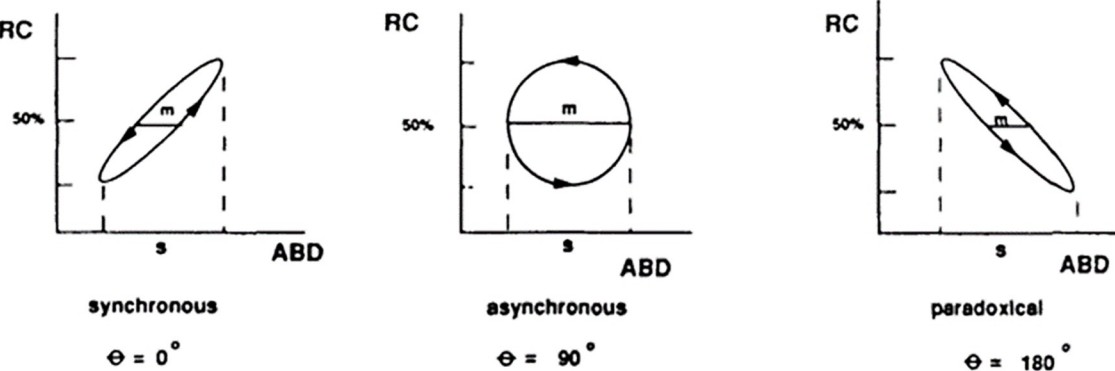

**Fig 1. Chest and abdomen plots during a single breath.** Reproduced from Sivan et al. [13]. Sample phase angles from chest (RC) and abdomen (ABD) data for three different single breaths: synchronous, asynchronous, and paradoxical. The variable m is the distance between the intercepts of the chest and abdomen loop with a horizontal line halfway between the minimum and maximum chest excursions, while the variable s is the maximum abdomen excursion. Phase angle is then calculated by taking the inverse sine of m/s, with the values always ranging between 0 and 180 degrees. Perfectly synchronous breathing is signified by a value of zero degrees.

Biomedical Institutional Review Board at The Ohio State University (approval number 2019H0183) and all participants provided written informed consent prior to beginning the study.

## Testing overview

Participants were brought to the laboratory on two separate days for testing. The testing days were separated by at least one week and participants were instructed to refrain from exercise within twenty-four hours of testing times to avoid possible impacts on testing day performance [23, 31–33]. All participants were instructed to wear comfortable running attire and the same self-provided shoes for both testing days.

The desired sample size for this study was estimated at fifteen participants, based on similar research involving exercise across multiple testing days that ultimately reported significant

**Table 1. Participant demographics.**

| Participant | Sex (M/F) | Age (years) | Height (cm) | Weight (kg) | Self-Reported Average km Run per Week | Days between Testing Visits |
|---|---|---|---|---|---|---|
| 1 | F | 19 | 165 | 56 | 14 | 7 |
| 2 | M | 22 | 183 | 79 | 10 | 19 |
| 3 | M | 24 | 189 | 82 | 27 | 39 |
| 4 | F | 28 | 173 | 67 | 10 | 14 |
| 5 | F | 41 | 164 | 71 | 32 | 8 |
| 6 | F | 22 | 170 | 61 | 8 | 30 |
| 7 | F | 35 | 168 | 70 | 24 | 35 |
| 8 | M | 22 | 183 | 91 | 26 | 7 |
| 9 | M | 21 | 175 | 64 | 26 | 16 |
| 10 | F | 24 | 163 | 50 | 19 | 7 |
| 11 | F | 23 | 170 | 78 | 32 | 7 |
| 12 | F | 20 | 170 | 60 | 13 | 14 |
| 13 | M | 31 | 177 | 73 | 16 | 12 |
| **Mean** | - | 25.5 | 173.0 | 69.4 | 19.8 | 16.5 |
| **SD** | - | 6.5 | 8.1 | 11.4 | 8.6 | 11.2 |

differences in performance due to dietary supplements [31, 33–36]. Of these studies, three specifically involved exercise to exhaustion [34–36] and the largest sample size was twelve [34].

On both testing days, participants completed a graded treadmill fatigue protocol to voluntary exhaustion [36]. On one of these testing days, real-time visual feedback on chest and abdomen expansion was provided to the participant during the run. Block randomization was used to ensure an even split of visual feedback on the first and second testing visits across the entire population. All running was performed on a Bertec split-belt instrumented treadmill (Bertec Corp., Columbus, OH).

## Visual feedback animations

A visual feedback interface was created in LabVIEW, consisting of two sets of images that were animated in sequence based on the voltage outputs from the RIP belts. Each set of images was independently controlled by its corresponding RIP belt. The chest was represented by an umbrella and the abdomen was represented by a rose, shown in Fig 2. The umbrella and rose were selected at the suggestion of a physical therapist, as they are common objects that can easily be interpreted by participants when wearing the RIP belts. A twenty-inch computer monitor was mounted to the front of the treadmill to provide feedback on the needed testing day, shown in Fig 3.

## Treadmill fatigue protocol

The treadmill fatigue protocol entailed setting the treadmill to a fixed three-degree incline with a starting speed of 2.25 m/s. Every minute, the treadmill would increase speed by 0.0833 m/s with an acceleration of 0.25 m/s$^2$. At the fifty-eighth second of every minute, a two-second beep would occur to notify the participant that the treadmill speed was about to increase. This protocol was selected based on similar research involving a running population undergoing treadmill testing to exhaustion [36] and to ensure that this wide-ranging population of runners all reached a true exhaustion point rather than simply not being able to move their legs fast enough to keep pace with the treadmill. An additional LabVIEW program was created to control the treadmill, which included automated speed changes and audible alerts. Prior to undergoing the treadmill fatigue protocol at a three-degree incline on each testing day, participants ran a five-minute warmup on the treadmill at a self-selected speed with no incline. The treadmill warmup speed was the same on both testing days for each participant. No visual feedback on breathing nor instruction for breathing were provided prior to the five-minute warmup. One-minute of rest followed this warmup.

## Breathing instructions for participants during the fatiguing run

Participants were provided with breathing information during the one-minute of rest after the warmup for the upcoming run to exhaustion. On both testing days, participants were told that the most efficient breathing entails the chest and abdomen expanding and contracting in synchrony and to try to exhibit this breathing pattern during the entire run. If the testing day did not include visual feedback, only this verbal explanation was provided. If the testing day included visual feedback, the visual feedback animations being displayed on the monitor were explained. Any questions the participant had pertaining to breathing or the animations were answered before beginning the fatigue protocol.

Participants were instructed to run to volitional exhaustion and to signal to the research team when this point was reached so the treadmill could be stopped.

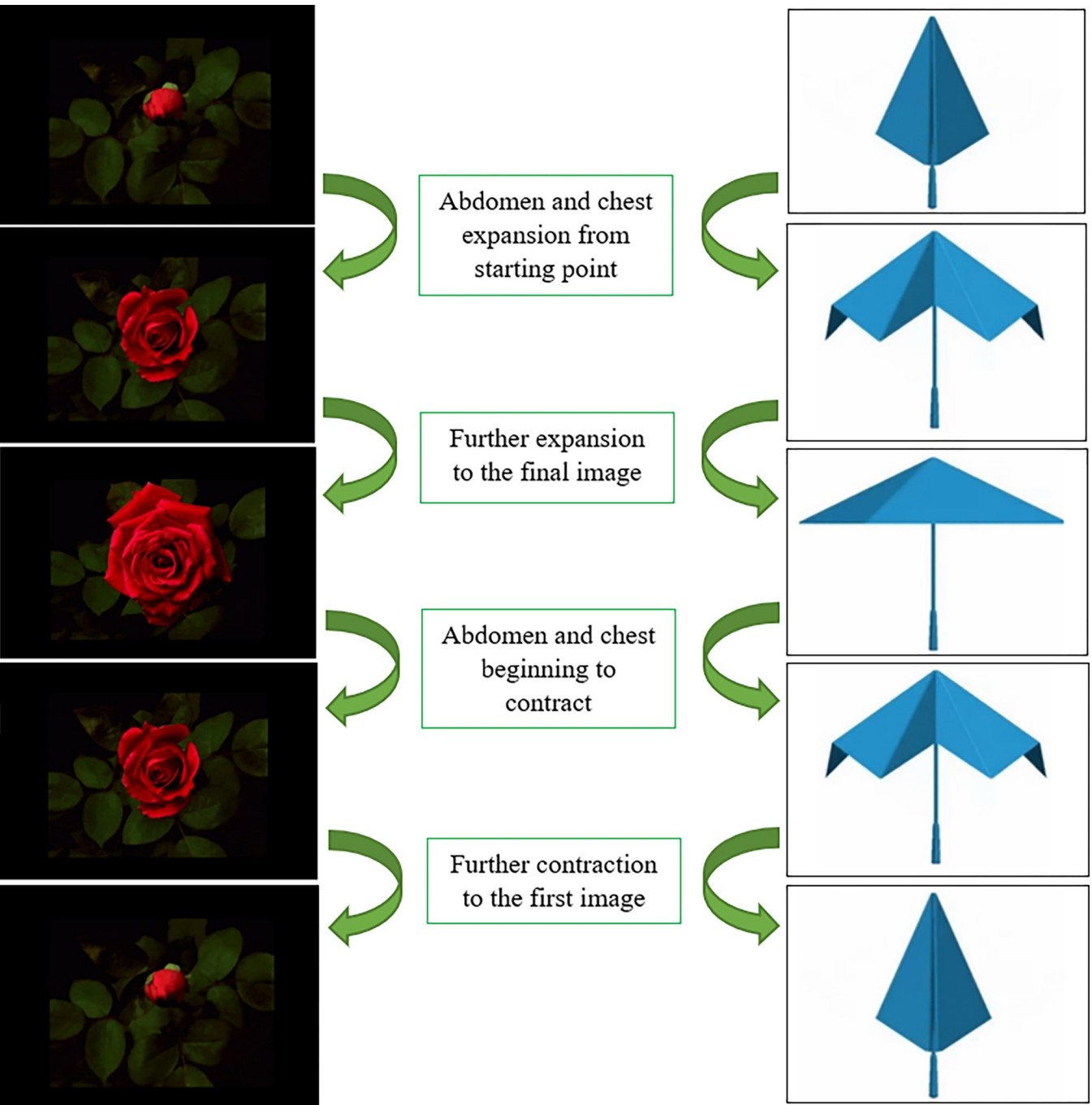

**Fig 2. Visual representation of the abdomen and chest animations.** The abdomen is represented by the rose and the chest is represented by the umbrella with each being independently controlled by the respective RIP belt. A change in the circumference of a belt results in a change in the voltage being output by the belt. The animations were programmed to change images based on these changes in voltage outputs. The actual animations contained 194 images for the rose and 156 images for the umbrella over the range of possible voltages.

## Equipment placement

Equipment placed on participants for all running tasks included two RIP belts placed around the chest and abdomen, a neoprene belt containing the data collection unit, a heart rate

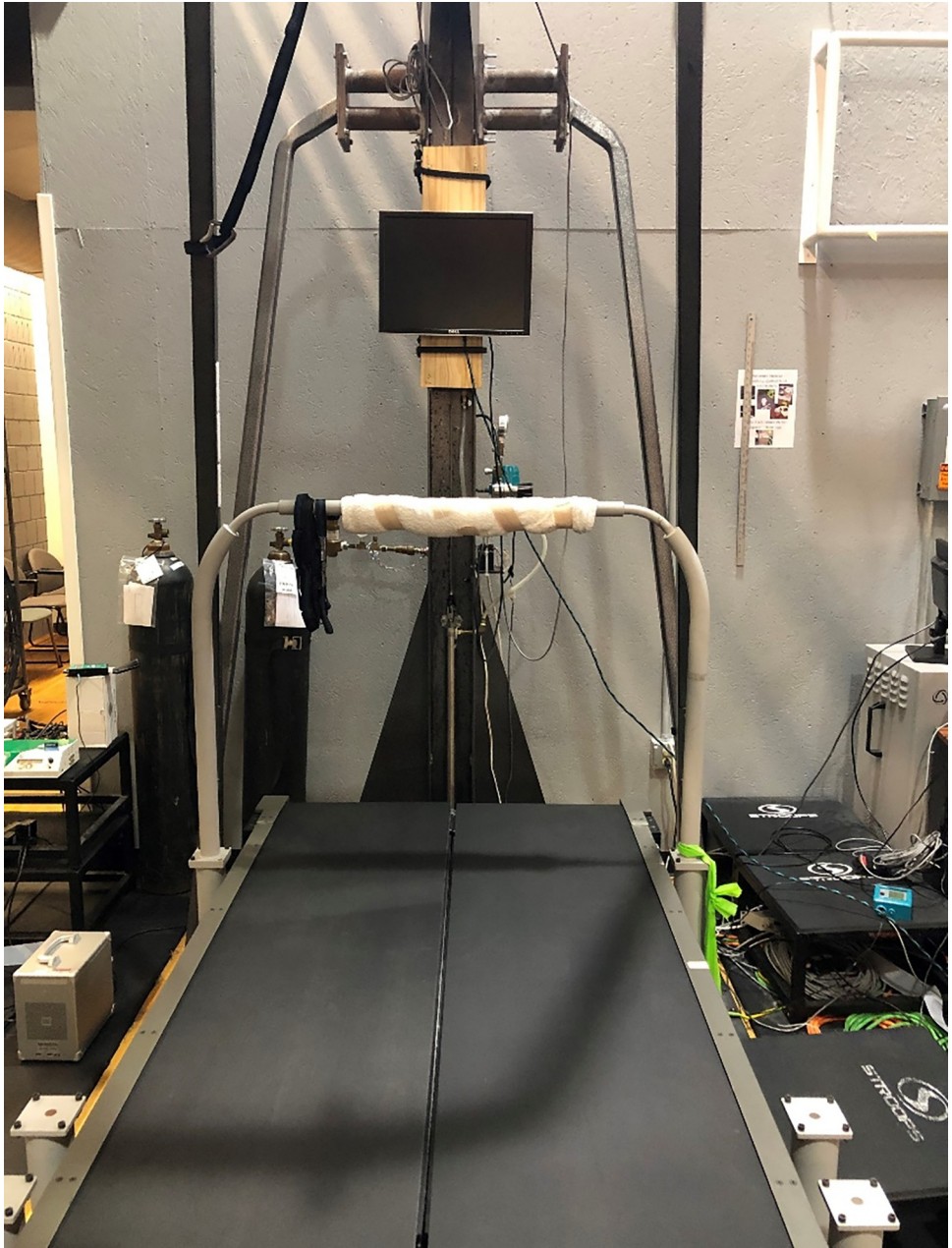

**Fig 3. Treadmill display.** The monitor mounted to the front of the treadmill that displayed the animations for real-time visual feedback.

monitor, and a safety harness, shown in Fig 4. The RIP belts were consistently placed on the participant with the chest belt at the level of the xyphoid process and below the pectoral muscles, and the abdomen belt at the level of the navel, as previous RIP research has prescribed [37, 38]. A neoprene belt with hook and loop fabric covering (CoreX Therapy, Perfect Practice, Inc.) was placed at the level of the waist to house the additional equipment needed for RIP, which included a BioRadio and two interface cables (Great Lakes NeuroTechnologies, Cleveland, OH). A Polar H7 heart rate monitor (Polar Electro Inc., Bethpage, NY) was placed on bare skin between the previously mentioned RIP belts. Each participant was fit with a custom-

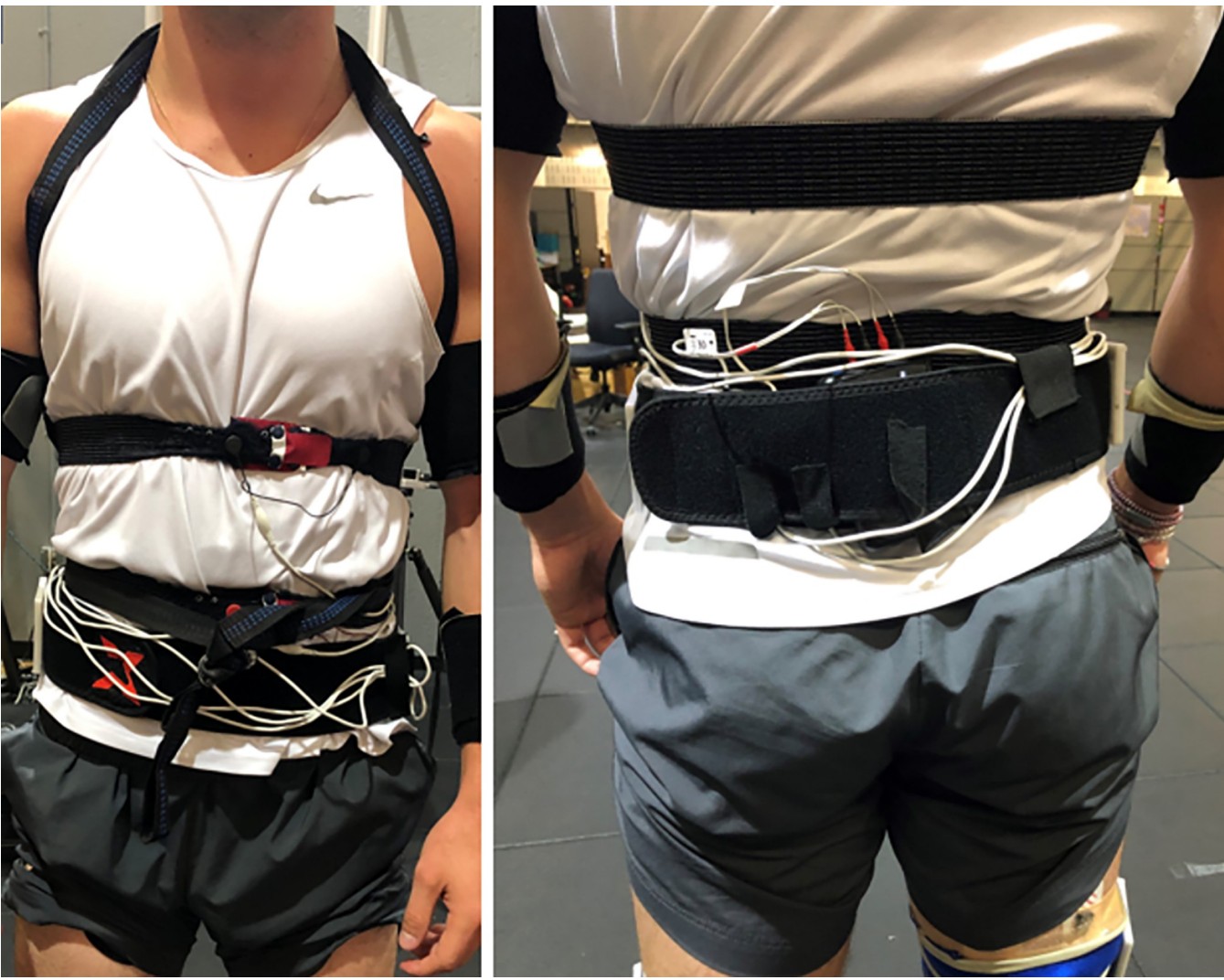

**Fig 4. Equipment worn by participants.** Anterior and posterior view of the equipment worn by participants while running. This equipment included two RIP belts, a heart rate monitor, a harness, and a neoprene belt. The neoprene belt housed the BioRadio, which was used to detect changes in voltage outputs of the belts, and additionally was where excess wiring from the RIP belts to the BioRadio was held down.

made harness [39] composed of two hammock straps (ENO Suspension, Asheville, NC) to prevent an injurious fall should they lose their footing.

## Data collection and processing

Breathing data were collected during the five-minute warmup and the fatigue protocol at 250 Hz. VivoSense version 3.3 (VivoSense, Newport Coast, CA) was used to filter and analyze the breathing data, which has been commonly used in exercise research with RIP [20–23]. Fig 5 illustrates the data processing steps automatically performed by VivoSense [14, 40, 41] to calculate the phase angle on a breath-by-breath basis for a typical breath in this dataset. A low-pass Butterworth filter (1.4 Hz cutoff) was applied to the breathing data, which has been shown to remove signal values that were generated from causes outside of breathing [23]. Voltages were converted to volumes for the chest and abdomen using a standard calibration algorithm, and the calibration weights the abdomen and chest bands with equal unit weighting.

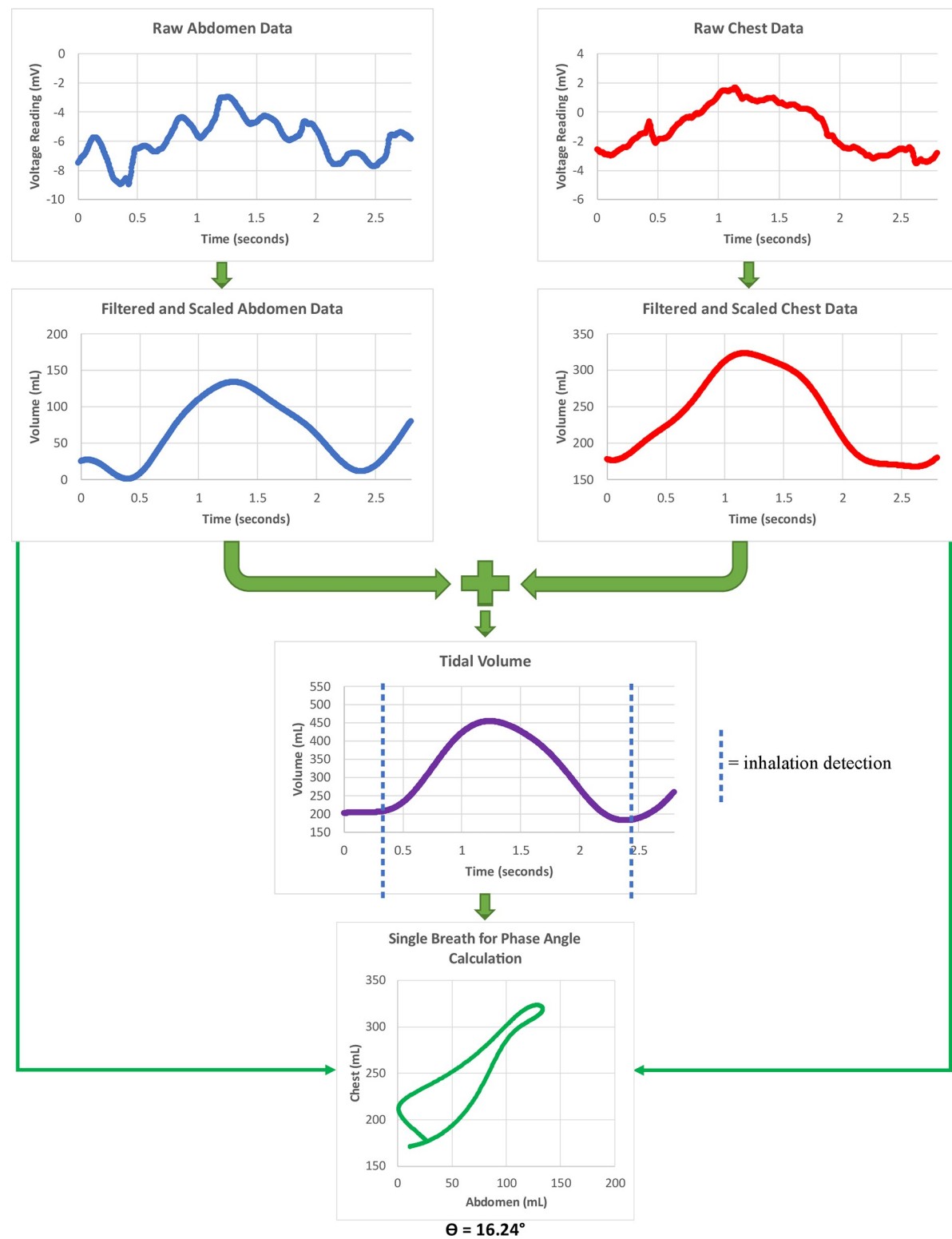

**Fig 5. The process used by VivoSense to calculate phase angle for each breath.** The entire process undergone by raw chest and abdomen data to calculate phase angle for a single breath.

The tidal volume is calculated as the sum of each filtered and scaled chest and abdomen point over time, thus forming another sinusoidal wave. Tidal, chest, and abdomen volumes are then proportionately scaled so that the average tidal volume over the entire testing session is 400mL. Detection of the beginning of inhalation for each breath was performed by (a) detecting the first incidence of a positive slope in the tidal volume, and (b) checking that the peak-to-trough change from the previous beginning of inhalation was greater than 48mL [41], also known as breath reversal detection. Once each breath, i.e. the interval from one beginning of inhalation to the next, had been identified, the phase angle for each breath could be calculated, as explained in Fig 1. These computations assume that chest and abdomen excursions are sinusoidal, so the filtered and calibrated chest and abdomen data form a loop (Fig 5, bottom). Breath-by-breath phase angle values were then averaged over the entire duration of the specific run of interest. Thus, three average phase angle values were calculated for each participant: warmup, run to exhaustion with visual feedback, and run to exhaustion without visual feedback.

Time to exhaustion was defined as the duration of the fatigue protocol from the beginning of the inclined run to volitional exhaustion. Heart rate beats per minute (BPM) data were output and recorded every minute during the fatigue protocol via the Polar Beat: Run & Fitness iOS application.

## Statistical analyses

For consistency across all variables of interest, univariate regression analyses were used to identify significant relationships for all continuous and logistic variables. Pairwise Pearson correlation coefficients were calculated for the variables of time to exhaustion, phase angle, warmup phase angle, age, sex, height, weight, a participant's kilometers run per week, days between testing periods, testing visit, and whether visual feedback was provided. Univariate linear regressions were performed for the dependent variables of time to exhaustion and phase angle using the variables listed previously as predictors. Statistical significance was set *a priori* at $\alpha = 0.05$. No correction for multiple comparisons was performed due to the exploratory nature of this study. All statistical analyses were performed in SPSS version 26 (IBM, Armonk, NY).

## Results

### Phase angle and time to exhaustion

Table 2 provides phase angle and time to exhaustion values for all participants during the runs to exhaustion on the inclined treadmill at increasing speed. The mean of the two warmup periods is reported as one warmup phase angle as the warmup is intended to provide a reference to a participant's uninstructed breathing pattern. Consistency of the phase angle during the runs to exhaustion was confirmed with no statistically significant difference between the phase angle during the first half of the run (mean 19.7˚ ±8.3˚) and the second half of the run (mean 19.0˚ ±8.6˚) (p = 0.676).

Fig 6 provides a visualization of each participant's performance with respect to phase angle and time to exhaustion. Each point was calculated by subtracting a participant's mean value from the treadmill fatigue protocol with no visual feedback from the treadmill fatigue protocol with visual feedback.

Time to exhaustion was not significantly predicted by phase angle (p = 0.853), nor having visual feedback provided (p = 1). Phase angle was not significantly predicted by having visual feedback provided (p = 0.667). The phase angle from the five-minute warmup significantly predicted phase angle during the runs to exhaustion (p = 0.045), meaning that those with a

**Table 2. Phase angle and time to exhaustion data for all participants.**

| Participant | Avg. Phase Angle from Warmup (Deg.) | Avg. Phase Angle with Visual Feedback (Deg.) | Avg. Phase Angle No Visual Feedback (Deg.) | Phase Angle Difference (Visual Feedback–No Visual Feedback) | Visual Feedback Time to Ex. (s) | No Visual Feedback Time to Ex. (s) | Time to Exhaustion Difference (Visual Feedback–No Visual Feedback) |
|---|---|---|---|---|---|---|---|
| 11 | 24.8 | 15.7 | 31.6 | -15.9 | 430.4 | 418.4 | 12.0 |
| 13 | 22.8 | 7.8 | 17.8 | -10.0 | 1025.9 | 960.8 | 65.1 |
| 12 | 13.4 | 8.3 | 17.6 | -9.2 | 378.3 | 362.5 | 15.8 |
| 9 | 14.5 | 20.1 | 29.1 | -9.0 | 1309.6 | 1337.5 | -27.9 |
| 10 | 14.7 | 11.6 | 12.3 | -0.6 | 542.9 | 540.9 | 1.9 |
| 1 | 38.6 | 13.8 | 14.0 | -0.2 | 488.9 | 516.2 | -27.3 |
| 5 | 19.0 | 20.3 | 20.2 | 0.1 | 323.8 | 300.9 | 22.9 |
| 7 | 12.7 | 11.9 | 11.1 | 0.9 | 369.6 | 397.8 | -28.2 |
| 2 | 20.1 | 26.7 | 25.3 | 1.4 | 766.4 | 805.6 | -39.2 |
| 6 | 23.6 | 31.9 | 29.7 | 2.1 | 362.0 | 356.0 | 6.0 |
| 4 | 36.8 | 31.4 | 25.4 | 6.0 | 373.8 | 484.7 | -110.9 |
| 3 | 11.8 | 21.1 | 14.3 | 6.9 | 563.1 | 521.0 | 42.1 |
| 8 | 29.4 | 22.8 | 12.4 | 10.4 | 729.9 | 661.8 | 68.1 |
| **Mean** | 21.7 | 18.7 | 20.1 | -1.3 | 589.6 | 589.6 | 0.0 |
| **SD** | 8.9 | 8.1 | 7.3 | 7.6 | 297.2 | 292.7 | 48.1 |

Sorted from greatest phase angle improvement to least. Shaded rows indicate a phase angle improvement of 9.0° or greater.

better mean phase angle during the five-minute warmup showed a better mean phase angle during the runs to exhaustion.

### Findings for other variables

Initial and final heart rates were not significantly different across testing visits (p = 0.552 and p = 0.802 respectively). Time to exhaustion was not significantly predicted by the order of testing visits (p = .919).

## Discussion

### Phase angle and time to exhaustion for the participant population

Phase angles were not significantly different across the entire group when comparing the five-minute warmup, visual feedback, and no visual feedback phase angle values. Moreover, the phase angle did not change significantly from start to end of the run to exhaustion. Our mean phase angle from the runs to exhaustion in this study of 19.4° (±7.7°) was similar to the mean phase angle of ten healthy adults undergoing an incremental treadmill protocol to exhaustion in another study (20.0°) [20].

No learning effect was observed with the treadmill fatigue protocol when going from the first testing day to the second, as time to exhaustion was not significantly different between the two (p = 0.919).

The observed time to exhaustion in our participants differed from the most similar study in the literature [36], but this difference is most likely due to the fact that Fraga et al. tested well-trained males while our participants were a mix of recreational athletes who run less than 32 kilometers per week. Other studies of time to exhaustion often used cycling instead of running, but we chose running due to its popularity at both a casual and competitive level. The study's hypothesis that an improvement in breathing could lead to an increased time to exhaustion

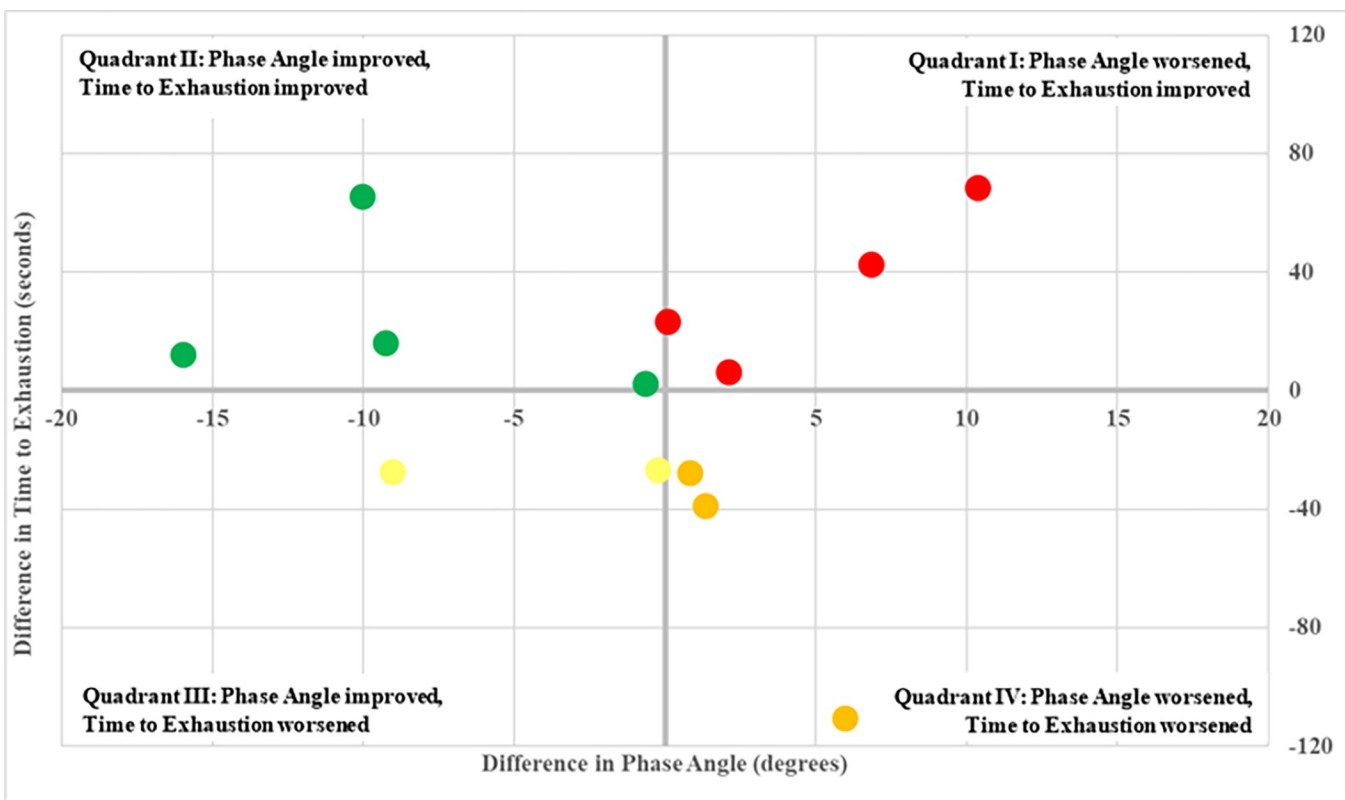

**Fig 6. Differences in time to exhaustion vs. differences in phase angle.** Each point was calculated by taking the values of phase angle and time to exhaustion from the visual feedback testing day and subtracting these same corresponding variable values from the testing day with no visual feedback for each participant. The colors of the points signify the quadrant that the point lies in, with red representing quadrant I, green for quadrant II, yellow for quadrant III, and orange for quadrant IV.

would be of interest to runners at any level, we chose to test recreational runners under the assumption that they were less likely to have previously learned any type of breathing technique for running than more experienced runners.

## Phase angle improvement and time to exhaustion

An improvement in phase angle did not significantly correlate with an increased time to exhaustion in this participant population. However, six participants had an improved phase angle during the run to exhaustion with visual feedback when compared to the run to exhaustion without visual feedback. Of these six participants, four increased time to exhaustion with visual feedback when compared to no visual feedback. Furthermore, there were four participants that had phase angle improvements of 9.0˚ or more, shown by the shaded rows in Table 2, three of whom had a positive time to exhaustion difference.

Analysis of these sub-groups suggests that improving phase angle could lead to an increase in time to exhaustion. However, not all participants improved phase angle when visual feedback was provided. This result could be due to several factors, but a primary one was likely the short familiarization period with the breathing technique of having the chest and abdomen expand and contract in synchrony. This breathing technique has been taught explicitly in previous research [28] and has been shown to have positive impacts on subjects when applied in various settings [9, 10] as it ultimately allows for the maximum amount of oxygen to enter the bloodstream with each breath [9–12]. Had greater steps been taken to familiarize participants

with the breathing technique, either through teaching or allowing participants to become familiarized with the visual feedback animations prior to running, there is potential that an improvement in phase angle and thus an increase in time to exhaustion could have been more prominent across all participants.

Furthermore, the mean phase angle from the runs to exhaustion being significantly predicted by the mean phase angle from the warmup shows the difficulty of changing breathing pattern with minimal time to practice or learn a new breathing method. It was hypothesized that the animations and instructions would be enough to cause significant change to the breathing pattern of a participant, but this level of familiarization was not enough to induce greater synchrony across all participants, further illustrating the need for practice with any type of new breathing technique.

## Future work and limitations

An improvement in phase angle did not ultimately lead to an increase in time to exhaustion across the entire study population. Looking specifically at participants who improved phase angle when visual feedback was provided, there is a possibility that time to exhaustion can be increased with an improved phase angle. An effective and efficient pulmonary system is important for endurance exercise performance, as several respiratory system-related mechanisms can impair oxygen delivery to locomotor muscles, such as arterial desaturation, and compromised cardiac output due to the demands of respiratory work [42]. Whether these factors played a role in our short-duration maximal exercise protocol and/or are related to phase angle remains to be tested. In the future, a more prolonged endurance effort with additional measures of breathing efficiency, such as $VE/CO_2$ could address additional potential impacts of the intervention.

Not all participants improved phase angle with visual feedback. As such, one obstacle is ensuring that all participants are breathing in the instructed manner of having the chest and abdomen expand and contract in synchrony. While subjective feedback from participants on the biofeedback system was not collected in a structured way in this study, participants were told to let the research team know if any breathing instructions were unclear or if they experienced any shortness of breath while running, and ultimately no participants reported any confusion on instructions or dyspnea during running. Thoraco-abdominal synchrony can be improved by manual breathing re-education methods or having a familiarization session for the participant with visual feedback. Given that this specific technique of visual feedback on breathing has not been used before, additional studies to gauge the time required for participants to fully learn this biofeedback technique are needed. With an established re-education method or familiarization session in place for future work, further time-series analyses can be done to observe phase angle patterns over time.

As shown in Table 2, the average time to exhaustion in this population of recreational runners was under 10 minutes, with six participants reaching volitional exhaustion in less than seven minutes. Studies with similar research objectives across multiple testing days [31, 33] utilized protocols that had participants exercise at a constant pace for an extended period which led to significant differences between the testing days. This further illustrates the potential limitation that a lower-intensity constant-pace run could lead to a larger difference observed between the two testing days. A constant-pace protocol tailored to each participant could further isolate the variable of breathing during maximal exertion exercise as participants would not need to meet the neuromuscular demands associated with increasing cadence throughout the protocol. As stated previously, given that the population in this study ran a wide-ranging number of kilometers per week, our protocol was selected to ensure that each individual could

reach a true exhaustion point rather than not being able to keep pace with the treadmill. Ensuring that participants reached a true exhaustion point in each testing session allowed us to confidently analyze the role that breathing played during maximal exertion exercise, and the role that breathing plays could potentially be greater in a constant-pace protocol. As with many physiological questions, the findings of this study are somewhat specific to the exercise protocol performed.

A primary limitation of this study was not having a testing day with a run to exhaustion without visual feedback or breathing instructions. This was not done for the primary reason of time commitment for participants, as well as the study ultimately being designed to isolate the impact that visual feedback had on participants. Our observation of time to exhaustion not being significantly predicted by the order of testing visits also suggests that the verbal or visual cues given on the first testing visit did not affect the results on the second testing visit. Additional limitations included the specific population tested and the fatigue protocol that participants were put through. It is possible that other running populations (such as novice runners of less than eight kilometers per week or more experienced runners of more than thirty-two kilometers per week) may have baseline phase angles that are inefficient and could see benefits from having visual feedback provided. Indirect calorimetry and spirometry testing on a separate day could be used in future studies to directly relate phase angle to other breathing and metabolic variables to determine the impact of visual feedback on running economy. It is also unknown to what extent the harness worn by participants impacted their breathing pattern, but it was assumed to have minimal influence. An additional limitation is the use of visual feedback for breathing instruction, which cannot be directly translated to overground running. However, given the lack of existing knowledge on whether breathing patterns during running can be changed at all, we believe that an approach providing clear, unambiguous feedback is a valuable first step to motivate the development of and assess the efficacy of other more portable types of feedback such as audio or haptic feedback [27]. Moreover, while many runners prefer field running, treadmill running is also very common, especially during wet or cold seasons. Therefore, we believe that further investigation of visual feedback to determine the upper limits of breathing training would be useful in parallel to developing non-visual breathing feedback approaches for overground running. Lastly, even with filtering applied to the raw voltages from the RIP bands based on the recommendations from previous studies [23] and a minimum tidal volume of 48mL used to minimize the number of erroneous breaths identified, over-identification of breaths could lead to noise in the estimation of the average phase angle. The dynamic nature of running with large vertical accelerations of the torso could have exacerbated this problem relative to studies of breathing during walking, sitting, or lying down.

## Conclusions

Visual feedback did not consistently lead to increased breathing efficiency or time to exhaustion across a population of healthy recreational runners. However, sub-analysis of participants with a phase angle improvement reveals the possibility that time to exhaustion can be increased if breathing improvements are made. Familiarization with any form of visual feedback or breathing technique should be of utmost importance in future studies.

## Acknowledgments

The authors would like to thank Maggie Abrams, DPT from The Ohio State University, and Bernard Tarver from Great Lakes NeuroTechnologies for their assistance and guidance with RIP technology. An additional thank you goes to Menglin Xu, PhD, statistician at The Ohio

State University, for her assistance with the statistical analysis. The authors also thank Nihit Tyagi for his assistance with data collection.

## Author Contributions

**Conceptualization:** Joseph A. Passafiume, Anne R. Crecelius, Ajit M. W. Chaudhari.

**Data curation:** Joseph A. Passafiume, Nelson A. Glover.

**Formal analysis:** Joseph A. Passafiume, Ajit M. W. Chaudhari.

**Investigation:** Joseph A. Passafiume, Nelson A. Glover, Ajit M. W. Chaudhari.

**Methodology:** Joseph A. Passafiume, Anne R. Crecelius, Ajit M. W. Chaudhari.

**Supervision:** Ajit M. W. Chaudhari.

**Validation:** Ajit M. W. Chaudhari.

**Writing – original draft:** Joseph A. Passafiume.

**Writing – review & editing:** Joseph A. Passafiume, Nelson A. Glover, Anne R. Crecelius, Ajit M. W. Chaudhari.

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
