## [Editor Report · Decision Letter 0]

29 Nov 2021

PONE-D-21-36124The influence of receiving real-time visual feedback on breathing during treadmill running to exhaustionPLOS ONE

Dear Dr. Passafiume,

Thank you for submitting your manuscript to PLOS ONE. After careful consideration, we feel that it has merit but does not fully meet PLOS ONE’s publication criteria as it currently stands. Therefore, we invite you to submit a revised version of the manuscript that addresses the points raised during the review process.Please provide addition details in the methods (outlined below) relating to the signal processing for the RIP sensors.

We look forward to receiving your revised manuscript.

Kind regards,

Caroline Sunderland

Academic Editor

PLOS ONE

Journal Requirements:

" ext-link-type="uri" xlink:type="simple">https://journals.plos.org/plosone/s/file?id=ba62/PLOSOne_formatting_sample_title_authors_affiliations.pdf"

Additional Editor Comments:

There are some concerns regarding the detail provided in the methods for using the RIP sensors. Therefore in the methods can you please provide:

1) Details regarding raw signal processing and breath onset detection

2) A raw data trace of the signal.
---

## [Author Response · Author response to Decision Letter 0]

31 Dec 2021

All responses are provided in the "Response to Reviewers" file.

---

## [Decision Letter · Decision Letter 1]

17 Jan 2022

PONE-D-21-36124R1The influence of receiving real-time visual feedback on breathing during treadmill running to exhaustionPLOS ONE

Dear Dr. Passafiume,

Thank you for submitting your manuscript to PLOS ONE. After careful consideration, we feel that it has merit but does not fully meet PLOS ONE’s publication criteria as it currently stands. Therefore, we invite you to submit a revised version of the manuscript that addresses the points raised during the review process.

We look forward to receiving your revised manuscript.

Kind regards,

Caroline Sunderland

Academic Editor

PLOS ONE

Journal Requirements:

Reviewers' comments:

Reviewer's Responses to Questions

**Comments to the Author**

1. If the authors have adequately addressed your comments raised in a previous round of review and you feel that this manuscript is now acceptable for publication, you may indicate that here to bypass the “Comments to the Author” section, enter your conflict of interest statement in the “Confidential to Editor” section, and submit your "Accept" recommendation.

Reviewer #1: (No Response)

Reviewer #2: All comments have been addressed

2. Is the manuscript technically sound, and do the data support the conclusions?

Reviewer #1: Yes

Reviewer #2: Yes

3. Has the statistical analysis been performed appropriately and rigorously? 

Reviewer #1: Yes

Reviewer #2: Yes

4. Have the authors made all data underlying the findings in their manuscript fully available?

Reviewer #1: Yes

Reviewer #2: No

5. Is the manuscript presented in an intelligible fashion and written in standard English?

Reviewer #1: Yes

Reviewer #2: Yes

6. Review Comments to the Author

Reviewer #1: I would like to commend the authors for this novel, important study seeking to improve breathing pattern during running. The exploration of such techniques in exercise is understudied. Furthermore, RIP sensors enable new possibilities when properly used. I stipulated that I would review this manuscript once some key methodological concerns were cleared up regarding the use of RIP to quantify the variable of interest in this study. Based on your edits, I am happy to review the rest of this manuscript.

• Line 31 “breathing synchronicity” – better to use one of the accepted terms “thoraco-abdominal coordination” or “thoraco-lumbar synchrony”

• Lines 41-42 is this reference correct?

• Lines 44-46 not sure this connects the previous and next sentences… can you provide info re: how breathing provides o2 and clears co2?

• Lines 49-50 this is a brief, bold statement supported by much theoretical knowledge but scarce experimental evidence. Citations 9-11 do not refer to T-L synchrony during exercise. Can you expand the concept of T-L synchrony independently before introducing RIP? I suggest you refer to the following: Bernardi, E., Pratali, L., Mandolesi, G., Spiridonova, M., Roi, G. S., Cogo, A. (2017). Thoraco-abdominal coordination and performance during uphill running at altitude. PLoS One, 12(3), e0174927.

• Lines 53-57 I suggest you check some updated references on RIP validity as there appear to be advances in this space in the last 10y. Specifically there are many recent studies on another commercial RIP device (Hexoskin): Harbour, E., Lasshofer, M., Genitrini, M., Schwameder, H. (2021). Enhanced Breathing Pattern Detection during Running Using Wearable Sensors. Sensors (Basel), 21(16).

• Lines 54-57 there are several studies examining the effect of breathing pattern interventions upon running. For a start, check out: Bahensky, P., Bunc, V., Malatova, R., Marko, D., Grosicki, G. J., Schuster, J. (2021). Impact of a Breathing Intervention on Engagement of Abdominal, Thoracic, and Subclavian Musculature during Exercise, a Randomized Trial. J Clin Med, 10(16) and Matsumoto, T., Masuda, T., Hotta, K., Shimizu, R., Ishii, A., Kutsuna, T., Yamamoto, K., Hara, M., Takahira, N., Matsunaga, A. (2011). Effects of prolonged expiration breathing on cardiopulmonary responses during incremental exercise. Respir. Physiol. Neurobiol., 178(2), 275-282.

• Lines 61-62 please populate this with normal values during exercise (Ref 16?).

• Lines 72-76 can you separate and clarify these points? Specifically, “breathing in healthy individuals is typically not optimally efficient” is a large claim likely difficult to substantiate with research. Maybe you mean to specify that healthy exercisers present with suboptimal T-L synchrony values and that improvement might also affect performance?

• Lines 80-86 can be easily be combined and made more concise.

• Lines 176-187 these lines all describe breath/flow reversal detection; can you explicitly label as such? Furthermore, if you have TV data, why not present this in results? Just a suggestion if there might be interesting comparisons.

• Lines 190-192 averages are good; do you have any other time-series analysis metrics that might show different aspects of breathing pattern? Propose variability of phase angle (as SD or CV) or slope (change from beginning to end). If you do not have these, include this caveat in your discussion.

• Lines 252-255 is this really the most similar study for comparison? Suggest to compare instead with another breathing strategy study using time to exhaustion: Wojta, D., Flores, X., Andres, F. (1987). 509: Effect of “Breathplay” on the Physiological Performance of Trained Cyclists. Med. Sci. Sports Exerc., 19(2), S85 or Dallam, G., McClaran, S., Cox, D., Foust, C. (2018). Effect of Nasal Versus Oral Breathing on Vo2max and Physiological Economy in Recreational Runners Following an Extended Period Spent Using Nasally Restricted Breathing. International Journal of Kinesiology and Sports Science, 6(2) or Vickery, R. L. (2008). The effect of breathing pattern retraining on performance in competitive cyclists [Auckland University of Technology].

• Lines 255-256 have you considered the unique constraints on breathing that running confers versus cycling and other sports? Suggest to check Elliott, A. D., Grace, F. (2010). An examination of exercise mode on ventilatory patterns during incremental exercise. Eur. J. Appl. Physiol., 110(3), 557-562.

• Lines 258-259 do you have any justification for this statement that experienced runners have a more efficient breathing pattern? Check out Salazar-Martinez, E., de Matos, T. R., Arrans, P., Santalla, A., Orellana, J. N. (2018). Ventilatory efficiency response is unaffected by fitness level, ergometer type, age or body mass index in male athletes. Biol. Sport, 35(4), 393-398. Moreover, elite athletes and females might be even more predisposed to respiratory limitations, especially at maximal intensity, as recently reviewed: Dempsey, J.A., La Gerche, A., and Hull, J.H. (2020). Is the healthy respiratory system built just right, overbuilt, or underbuilt to meet the demands imposed by exercise? J Appl Physiol (1985) 129, 1235-1256.

• Lines 260-266 does not strengthen this discussion for me. If you wish to retain, please expand upon why this protocol selection is important: is it better simply for intra-subject equivalence? Is it superior for eliciting respiratory limitations?

• Lines 279-281 can you put this in context with the results of Bernardi, 2017 as mentioned above? What does T-L synchrony have to do with insp/exp airflow and the work of breathing?

• Lines 281-284 do you have any idea what amount of time is needed to optimize this familiarization? There should be some contextual comparison in biofeedback studies and some of the previous references in learning breathing techniques.

• Lines 285-288 clunky; reword

• Can you add some comments to the discussion re: field vs. laboratory running? The implication here is that this breath retraining might have long-term effects on breathing during running. Can you also make a comment on feedback learning/self-awareness of T-L synchrony if expected to carry into field running?

• Lines 297-302 a large limitation I see is missing subjective feedback such as dyspnea, perceived efficacy/difficulty (to breathing instructions), and cognitive load. Please comment.

• Line 300-301 first part of sentence says nothing; how about “manual breathing re-education methods” or the like?

• Lines 303-306 also consider the limitations of condition randomization – the order of verbal vs visual cues might have affected the instruction adherence of the next session.

• Can you sharpen figure 5? Not only it is blurry, but also unclear. Specifically, I am guessing that there should not be an arrow between the summed tidal volume and phase angle graphs.

Reviewer #2: Dear Authors,

Thank you for addressing all the comments. Your revisions substantially improved the content of the manuscript. Keep up the good work.

Best regards,

Reviewer

7. PLOS authors have the option to publish the peer review history of their article (what does this mean?). If published, this will include your full peer review and any attached files.

Reviewer #1: No

Reviewer #2: **Yes: **Seyed Hamed Mousavi

---

## [Author Response · Author response to Decision Letter 1]

31 Jan 2022

All responses are provided in the "Response to Reviewers" file.

---

## [Decision Letter · Decision Letter 2]

17 Mar 2022

PONE-D-21-36124R2The influence of receiving real-time visual feedback on breathing during treadmill running to exhaustionPLOS ONE

Dear Dr. Passafiume,

Thank you for submitting your manuscript to PLOS ONE. After careful consideration, we feel that it has merit but does not fully meet PLOS ONE’s publication criteria as it currently stands. Therefore, we invite you to submit a revised version of the manuscript that addresses the points raised during the review process.

Please respond to the reviewers remaining comments, addressing them in a point by point manner.

We look forward to receiving your revised manuscript.

Kind regards,

Caroline Sunderland

Academic Editor

PLOS ONE

Journal Requirements:

Reviewers' comments:

Reviewer's Responses to Questions

**Comments to the Author**

1. If the authors have adequately addressed your comments raised in a previous round of review and you feel that this manuscript is now acceptable for publication, you may indicate that here to bypass the “Comments to the Author” section, enter your conflict of interest statement in the “Confidential to Editor” section, and submit your "Accept" recommendation.

Reviewer #1: (No Response)

2. Is the manuscript technically sound, and do the data support the conclusions?

Reviewer #1: Yes

3. Has the statistical analysis been performed appropriately and rigorously? 

Reviewer #1: I Don't Know

4. Have the authors made all data underlying the findings in their manuscript fully available?

Reviewer #1: No

5. Is the manuscript presented in an intelligible fashion and written in standard English?

Reviewer #1: Yes

6. Review Comments to the Author

Reviewer #1: (No Response)

7. PLOS authors have the option to publish the peer review history of their article (what does this mean?). If published, this will include your full peer review and any attached files.

Reviewer #1: No

---

## [Author Response · Author response to Decision Letter 2]

11 May 2022

All responses are provided in the "Response to Reviewers" file.

---

## [Decision Letter · Decision Letter 3]

9 Jun 2022

The influence of receiving real-time visual feedback on breathing during treadmill running to exhaustion

PONE-D-21-36124R3

Dear Dr. Passafiume,

We’re pleased to inform you that your manuscript has been judged scientifically suitable for publication and will be formally accepted for publication once it meets all outstanding technical requirements.

Kind regards,

Caroline Sunderland

Academic Editor

PLOS ONE

Additional Editor Comments (optional):

Reviewers' comments:

Reviewer's Responses to Questions

**Comments to the Author**

1. If the authors have adequately addressed your comments raised in a previous round of review and you feel that this manuscript is now acceptable for publication, you may indicate that here to bypass the “Comments to the Author” section, enter your conflict of interest statement in the “Confidential to Editor” section, and submit your "Accept" recommendation.

Reviewer #1: All comments have been addressed

2. Is the manuscript technically sound, and do the data support the conclusions?

Reviewer #1: Yes

3. Has the statistical analysis been performed appropriately and rigorously? 

Reviewer #1: Yes

4. Have the authors made all data underlying the findings in their manuscript fully available?

Reviewer #1: No

5. Is the manuscript presented in an intelligible fashion and written in standard English?

Reviewer #1: Yes

6. Review Comments to the Author

Reviewer #1: (No Response)

7. PLOS authors have the option to publish the peer review history of their article (what does this mean?). If published, this will include your full peer review and any attached files.

Reviewer #1: No

---

## [Editor Report · Acceptance letter]

27 Jun 2022

PONE-D-21-36124R3 

The influence of receiving real-time visual feedback on breathing during treadmill running to exhaustion 

Dear Dr. Passafiume:

I'm pleased to inform you that your manuscript has been deemed suitable for publication in PLOS ONE. Congratulations! Your manuscript is now with our production department. 

Kind regards, 

on behalf of

Dr. Caroline Sunderland 

Academic Editor

PLOS ONE